# Conventional and Complementary Medicine Health Care Practitioners’ Perspectives on Interprofessional Communication: A Qualitative Rapid Review

**DOI:** 10.3390/medicina55100650

**Published:** 2019-09-27

**Authors:** Janet Nguyen, Lorraine Smith, Jennifer Hunter, Joanna E. Harnett

**Affiliations:** 1Faculty of Medicine and Health, School of Pharmacy, The University of Sydney, Sydney, NSW 2006, Australia; jngu0475@uni.sydney.edu.au (J.N.); lorraine.smith@sydney.edu.au (L.S.); 2Faculty of Medicine and Health, School of Public Health, The University of Sydney, Sydney, NSW 2006, Australia; jennifer.hunter@westernsydney.edu.au; 3NICM Health Research Institute, Western Sydney University, Penrith, NSW 2751, Australia

**Keywords:** interprofessional relations, health communication, pharmacists, nurses, physicians, complementary medicine, health personnel

## Abstract

*Background and Objectives:* People have multi-faceted health care needs and consult a diverse range of health care practitioners (HCP) from both the conventional and complementary medicine healthcare sectors. The effective communication between HCP and with patients are obvious requisites to coordinating multidisciplinary care and shared decision making. Further, miscommunication is a leading cause of patient harm and is associated with reduced patient satisfaction, health literacy, treatment compliance and quality of life. In conventional healthcare settings, the differences in professional hierarchy, training, communication styles and culture are recognised communication barriers. Less is known about interprofessional communication (IPC) that includes traditional and complementary medicine (TCM) HCP. This review aims to summarise the experiences and perceptions of conventional and complementary HCP and identify factors that influence IPC. *Methods:* A qualitative rapid literature review was conducted. Six databases were searched to identify original research and systematic reviews published since 2009 and in English. Excluded were articles reporting original research outside of Australia that did not include TCM-HCP, already cited in a systematic review, or of low quality with a score of less than three on a critical appraisal skills programme (CASP) checklist. A thematic analysis of included studies was used to identify and explore important and recurring themes. *Results:* From the conducted searches, 18 articles were included, 11 of which reported data on complementary HCP and seven were literature reviews. Four key themes were identified that impact IPC: medical dominance, clarity of HCP roles, a shared vision, and education and training. *Conclusion:* IPC within and between conventional and complementary HCP is impacted by interrelated factors. A diverse range of initiatives that facilitate interprofessional learning and collaboration are required to facilitate IPC and help overcome medical dominance and interprofessional cultural divides.

## 1. Introduction

Interprofessional communication (IPC) between health care practitioners (HCP), services and patients play a pivotal role in the delivery of safe and effective, patient-centred care. Effective IPC has direct and positive implications in reducing morbidity and costs associated with communication failures and through improving the well-being and satisfaction of patients [1]. Indeed, communication failures between health care practitioners (HCP), non-clinical staff and patients are a leading cause of inadvertent patient harm [2]. The failure of HCP to communicate effectively within a multidisciplinary setting has been associated with reduced patient satisfaction, health literacy, treatment compliance and quality of life [3,4]. Patients are often left feeling caught between differing clinical opinions and recommendations [5].

The known elements of effective IPC include building professional relationships where there is continuity of care and a respect for the skills and expertise of the other HCP involved in the care of the patient [6]. A considerable amount of the IPC research arises from the imperative to improve communication skills and collaboration between medical doctors and nurses [4]. The substantial challenges to effective IPC include the different training, education, language and roles of medical doctors and nurses that are reinforced by a hierarchical health service [4]. These challenges are further amplified by the complexity of co-ordinating multidisciplinary care that increasingly involves a wide range of HCP, such as pharmacists [7] and allied health [5], as well as traditional and complementary medicine (TCM) [8]. For the purpose of this paper, TCM-HCP refers to practitioners of traditional and complementary medicine that is defined by the WHO as alternative medical systems that are not fully integrated into a country’s dominant, conventional health care system [8].

While the merits and importance of effective IPC between conventional HCP have been reported [4,7], a systematic review of the literature that includes TCM-HCP has not been performed. Given the high uptake of TCM that is often used alongside conventional healthcare [8], a broader systematic review is warranted. Therefore, the aim of this review is to investigate and synthesise the current literature in a timely manner that reports conventional and TCM HCP experiences with, and perceptions of IPC, and identify factors that influence IPC. The country of interest was Australia.

## 2. Methods

A qualitative literature review [9] using a rapid review approach [10] was conducted to retrieve relevant data and conduct a thematic analysis. The objective of this study was to thematically analyse the data obtained from a range of qualitative studies in a short time frame. Rapid reviews where the systematic review methods are restricted, are increasingly being used to answer questions in a timely manner that is fit-for-purpose [10,11,12]. The data from relevant studies were identified, evaluated and synthesised using a rapid review methodology [10]. In this instance, the methods were streamlined to generate meaningful results within a 3-month timeline to be used by the authors and their associates to inform undergraduate, postgraduate and continuing professional development activities in Australia. To this end, the scope of eligible studies was restricted by date, language, country and types of HCP, and the data extraction included the results of literature reviews.

### 2.1. Search Strategy

Two authors agreed on the search terms and the inclusion/exclusion criteria prior to one author conducting the search. Six databases were searched: Cumulative Index to Nursing and Allied Health Literature (CINAHL), Embase, Scopus, Medline, PubMed and Web of Science to identify original research and literature reviews written in English and published within the last ten years from January 2009 to March 2019. These databases report on communication between a broad range of HCP with a focus on Australian medical doctors/physicians, nurses, pharmacists and TCM-HCP. Excluded were articles that did not consider at least one of the HCP of interest, original research that was cited in eligible literature reviews, and original research that either did not include data about Australian HCP or TCM-HCP from any country. The timeframe was chosen for relevance in reflecting IP practices within the last 10 years. The literature reviews are typically included in rapid reviews to capture the perspectives from other authors who have also explored the aspects of this area in depth. The search strategy from Medline is presented in Figure 1. Additional articles were identified by searching Google Scholar and by citation chaining. After removing duplicate articles, the title and abstract were screened by two authors for appropriateness. The shortlisted full-text articles were reviewed in depth and the relevant data was extracted based on consensus of three authors. The authors of this review come from diverse clinical backgrounds representing the fields of medicine, psychology, integrative medicine, pharmacy and naturopathy. A minimum of three authors were involved in coding and the interpretation of the themes and subthemes and in reaching consensus. The critical appraisal skills programme (CASP) qualitative checklists [13] were used to rate the quality of the studies included in this review [14]. The CASP checklist comprised of 11 questions with options, yes, can’t tell and no. The checklist enables a systematic quality evaluation of qualitative studies. The studies were graded as good (more than six yes scores), moderate (between three-six yes scores) or poor (less than three yes scores). The studies classified as poor were excluded. The studies graded as good consisted of relevant topics on barriers and facilitators of interprofessional communication with a focus on conventional and TCM-HCP with rigorous execution. The studies graded as moderate consisted of relevant topics and rigorous to a lesser degree whereas the studies graded as poor did not address the relevant topics.

### 2.2. Data Analysis

The data analysis began by extracting data reporting the publication year, aims, study method, sample characteristics and key findings [15]. The key findings were extracted and deduced by one author (Janet Nguyen) and the validity was assured through validation with another author (Joanna Harnett). The data was manually extracted for the thematic analysis with the aim of supporting the objective rationale and interpretive thought processes. The Braun and Clark’s (2006) [16] six methods approach for literature synthesis was utilised. Step 1: Data familiarisation, was initiated through the identification of key concepts within the primary qualitative articles by one author. Step 2: Code generation, involved systematically collating and categorising relevant information regarding IPC. Step 3: Theme exploration, involved collating data into potential sub-themes and arranging common sub-themes from the array of studies accordingly with a focus on exploring the commonalities and differences in the perspectives of conventional and TCM-HCP. Step 4: Theme review, involved confirming the themes were in synergy with the extracts and the data set presented and generating a thematic map. Step 5: Theme defining and naming, involved refining the descriptive and analytical themes. Step 6: Report construction, involved a final analysis of the data, relaying results back to the study question and ensuring consensus between the three reviewers was met through discussion [16]. The discussion allowed critical review, data interpretation, clarification and resolution of disparities.

## 3. Results

The literature search yielded 457 articles with additional records retrieved from supplementary sources (Figure 2). The title and abstract screen shortlisted 46 articles from which 26 full text articles were critically appraised and 18 were included in the review. 

Represented in the included articles were the perspectives of IPC from the viewpoints of the following HCP: medical doctors (*n* = 11), nurses (*n* = 4), pharmacists (*n* = 4), TCM-HCP (*n* = 6) and other HCP such as allied HCP, health service managers (HSM), academics, counsellors, hospital administrators, board members, mental health practitioners, receptionists and key stakeholders (*n* = 7). Allied HCP refers to trained practitioners contributing to a patient’s diagnosis, management and prevention such as physiotherapists, dietetics and speech pathologists [18]. The quality of most studies was rated as good (*n* = 14). The study methods consisted of literature reviews (*n =* 8), surveys or questionnaires (*n* = 3), semi-structured interviews (*n* = 2) and mixed methods (*n* = 5). An overview of included studies is summarised in Table 1.

Four key themes were identified: (1) medical dominance, (2) clarity of HCP roles, (3) a shared vision, and (4) education and training (Figure 3). The four themes were further categorised into 15 interrelated sub-themes. An overview of the key themes and sub-themes is summarised in Table 2. The indicative quotes directly extracted from the articles are derived from the perspective and experiences of both conventional and TCM-HCP. After identifying recurrent themes, the credibility of the findings was discussed among three authors until an agreement was met about the key themes.

## 4. Discussion

The key themes identified in this review including medical dominance, clarifying the roles of HCP, shared visions and education and training are worthy of discussion. There are several review papers that have summarised different aspects of IPC [4,7,22,33], including facilitators and barriers to IPC between conventional HCP. To the best of the authors’ knowledge, this is the first qualitative literature review that combines research on IPC between both conventional and TCM-HCP. Further, this literature review has explored beyond facilitators and barriers of IPC and hence provides a comprehensive overview from the perspective and experiences of HCP. Through the systematic evaluation and collection of published data, relevant information has been rigorously extracted. The results from this qualitative, rapid review provide an in-depth overview of the enablers and barriers to effective IPC. Through a comprehensive search on relevant literature, this review is the first to include the perspectives and experiences of both conventional and TCM-HCP. 

### 4.1. Medical Dominance

Medical dominance, is defined by the Health Sociology Review as the ethos of the medical profession “exerting sovereign power over other professions such as nursing” [34]. Medical dominance was found to be a prohibiting, restrictive and limiting factor of effective IPC and was a prominent and recurrent theme amongst the studies included in this review [1,7,19,21,22,23,26,29,30,31,32]. Both conventional and TCM-HCP are negatively impacted by the dominant position of medical doctors that in turn, governs healthcare practice and service delivery, trust between HCP, medico-legal implications, financial remunerations and policy. The dominance of the scientific approach and evidence-based medicine is identified as a significant barrier to IPC with TCM-HCP. 

Nurses [1], pharmacists [7] and TCM-HCP [19] state that the medical doctors’ dominant position undermines their roles and has a restrictive effect that limits their contribution to patient care [21]. Within the hierarchy of medical practice, medical doctors typically assume a leadership position in overseeing patient care [26] that is reinforced by the organisation, funding and policies of the dominant healthcare system [23]. Within this medically dominant paradigm, IPC is limited by behaviours, such as a reluctance by medical doctors to disclose their rationale for treatment approaches to other HCP [1,26]. The behaviour is hypothesised to be an attempt by medical doctors to preserve their practitioner autonomy and leadership position [1]. In addition to practitioner autonomy, there is some evidence to suggest that medical doctors doubt other HCP competencies [4]. For example, one study suggested that doctors doubt pharmacists’ competence with delivering a broader range of healthcare services [7]. Confounding these factors are doctors’ concerns about confidentiality issues when sharing mutual patient information [30]. Therefore, medical doctors may limit their communication and referrals to other medical doctors and allied HCP due to potential medico-legal issues associated with sharing information and making referrals [4,30]. The factors associated with privacy and medico-legal responsibilities may inadvertently be limiting and restricting IPC, and therefore blocking effective multi-disciplinary care. 

It is suggested that improvements in IPC lies within the hands of policymakers, HCP and other key stakeholders (including patients) to coordinate the development and execution of policies that facilitate fluid and effective communication channels between medical doctors and other HCP [19]. The guiding principles that inform policy through defining and streamlining communication channels between HCP and other stakeholders could potentially disarm the negative effects created by medical dominance and provide a well-defined framework for multi-disciplinary patient-centred care [26].

### 4.2. Clarity of HCP Roles

According to the Canadian Interprofessional Health Collaborative, role clarification encompasses “learners/practitioners [who] understand their own role and the roles of those in other professions” [35] and that this has a powerful influence on IPC. The second theme identified in this review supports this view and further highlights the importance of understanding and respecting the roles and expertise of other HCP involved in patient care as a basis to initiating and executing effective IPC [1,6,7,20,21,22,26,30,31].

In the absence of a clear understanding and a lack of definition about the roles and expertise of other HCP, particularly TCM-HCP report being disempowered and their skills and knowledge under-utilised [21]. TCM-HCP report being excluded from communication related to patient care and their communication attempts are dismissed and poorly received by other HCP [25]. Other HCP such as nurses’ and pharmacists’ roles and responsibilities are additionally under-utilised by other HCP due to a poor understanding of their skills and expertise [1,7]. This has a bidirectional effect on communication as once one discipline of health care is not fully cognisant of the competencies [7] of another discipline, IPC is restricted and results in low levels of respect [7] and a sense of being undervalued [19] by other HCP. In addition, when another HCP role is poorly understood, communication is misinterpreted, and unjustified role dissention emerges [7,21].

The innovative strategies that encourage collaboration include increasing the opportunities for informal communication, such as face-to-face contact and formal communication which includes shared access to a clinic’s intranet [22]. However, in order for the healthcare system to embrace and advance opportunities for collaborative care through IPC, an understanding of the respective HCP expertise and skills and roles are required. The strategies that encourage such understanding would ultimately bring awareness, value and respect about the roles HCP can play to improve the healthcare delivery to patients. It could be extrapolated that through a clearer understanding and respect of other HCP roles, there would be an increase in referral and broader utilisation of the full spectrum of services thus placing an increased demand on the need for IPC tools. 

### 4.3. Shared Visions

A holistic and patient centred approach is facilitated when all the HCP involved in a patient’s care, along with the patient [4], share a similar vision for management strategies and outcomes [27]. Consequently, a shared vision is an important factor in facilitating and enabling effective IPC [6,7,21,22,27,30,32]. There is also an array of formal communication approaches which can further support HCP collaboration, shared decision-making and IPC [6,7,21,22,27,30,32].

A collaborative approach to exchanging information between HCP [25] and developing harmonious partnerships [1] can help HCP and patients identify common goals and a shared vision that in turn promotes cohesive healthcare delivery and IPC. The barriers to a shared vision and IPC include a disregard for the value of other HCP [19], perceived threats to income [1,26], and the belief that collaboration and IPC are not always necessary or possible, particularly when the interventions are completely different as is often the case when comparing conventional medicine and TCM [19].

The organisational, formalised approaches, such as clinical meetings, case reviews and conferences [1,4,24,26,27,30], are suggested as effective strategies to overcoming HCP resistance and to addressing any concerns about collaboration and teamwork [26]. The implementation of an agreed common language and communication style can further support formal approaches by helping diverse HCP to find common ground in their philosophies of care and develop common understandings of the scope of practice and clinical expertise [21,22,30].

The co-location of a multidisciplinary team within a health facility is another facilitator of IPC. Regular meetings enable HCP to become more familiar with the roles of other HCP and their vision for patient care [27]. Case conferences and practice meetings where patient data is shared, and joint decisions are made help to optimise management plans for the patient’s benefit [1,26]. However, the practicalities of time and travel often limit face-to-face meetings and therefore IPC. To support the development of a shared vision and IPC outside of a co-location environment, professional organisational initiatives and directives that prioritise time and space for team meetings that facilitate interprofessional learning are proposed [32].

### 4.4. Education and Training

Lifelong learning through practice and continuing education and training are central to the further development of knowledge, expertise and the skills that are required to be current and relevant to patient care. A predominant theme in the studies included in this review was the importance of education and training as fundamental strategies for improving IPC [1,6,7,19,20,21,25,30,31,32]. Education is needed to support appropriate formal and informal IPC and education, be it in person, on paper or electronically, and to improve HCP knowledge about other healthcare practices, philosophies and paradigms. The standardisation of the language used by HCP can further enhance IPC and thus promote understanding of the care that mutual patients are receiving [25,30].

Medical doctors can have negative beliefs about the standard of education of other HCP that impedes effective and respectful IPC [1]. TCM-HCP in particular are often viewed as research illiterate with a limited ability to contribute or engage in a robust discussion within an evidence-based medical paradigm [25]. To overcome this barrier, the inclusion of critical appraisal skills and research methodology within TCM-HCP education is viewed as a pre-requisite for conventional HCP accepting and including TCM-HCP in IPC [25].

In addition to improving research literacy, TCM-HCP in particular, use terminology and language unique to their traditional medicine system when describing clinical presentations, diagnosis and management. The different terminology and language that are typically not understood and potentially misinterpreted by both conventional and traditional HCP act as a further barrier to initiating and reciprocating IPC. Supper et al. [30] highlighted the positive effects of standardising a language set between health care disciplines to facilitate knowledge sharing and creating problem solving and promoting holistic health care. In addition to the desired effect of optimising and improving patient care and outcomes, such education and training has been shown to improve self-confidence and promote the respective professions’ roles thus enabling greater role-equity [21].

The initiatives in general practitioner education suggest that evidence-based factsheets on complementary approaches to health care are an effective strategy to improving a medical doctor’s knowledge of TCM and facilitating IPC [24]. Within conventional healthcare, the implementation of training programs focused on a medical doctors’ knowledge of particular therapeutic areas have been proposed to facilitate engagement with pharmacists [20] and promote a broader perspective and approach to medicine use that ultimately improves patient outcomes. This work further supports the premise that education about other HCP is an effective strategy for improving interprofessional collaborations and facilitating IPC.

Despite the execution of this review, there are several limitations that require consideration. Whilst a broad number of databases were searched, key publications may have been missed due to the chosen search terms. Focusing on only four different HCP—medical doctors, nurses, pharmacists and TCM-HCP—may have resulted in other important findings or nuances relevant to other allied health providers and original research not from Australia being missed. Notwithstanding this, five of the studies, including one systematic review included the perspectives of other types of conventional HCP, the systematic reviews included original research from numerous countries, and data saturation of the key themes were reached. The decision to focus on Australia helped to create a homogenous dataset with improved generalisability for a single national health system. The results of this study provide opportunities for cross-cultural comparison research in the future. Employing the method of thematic analysis may have led to the disregard of important data due to the inherent biases from the authors that could further influence the selection and interpretation of the emerging themes. The authors’ preference to use manual methods rather than a qualitative analysis program may have introduced errors and miscoding. The strategies employed to minimise these risks included constant cross-referencing with the original manuscripts that was independently performed by two authors. 

## 5. Conclusions

In summary, this review has identified four key themes that influence IPC, with implications for HCP, patients and the safety and quality of healthcare delivery. Due to the pervading dominant role of medical doctors in health care and society in general, overcoming the limitations associated with medical dominance will remain an ongoing challenge for effective IPC. The education and training initiatives that facilitate interprofessional learning and collaboration and are cognisant with an interdisciplinary patient-centred approach are an important part of this process.

## Figures and Tables

**Figure 1 medicina-55-00650-f001:**
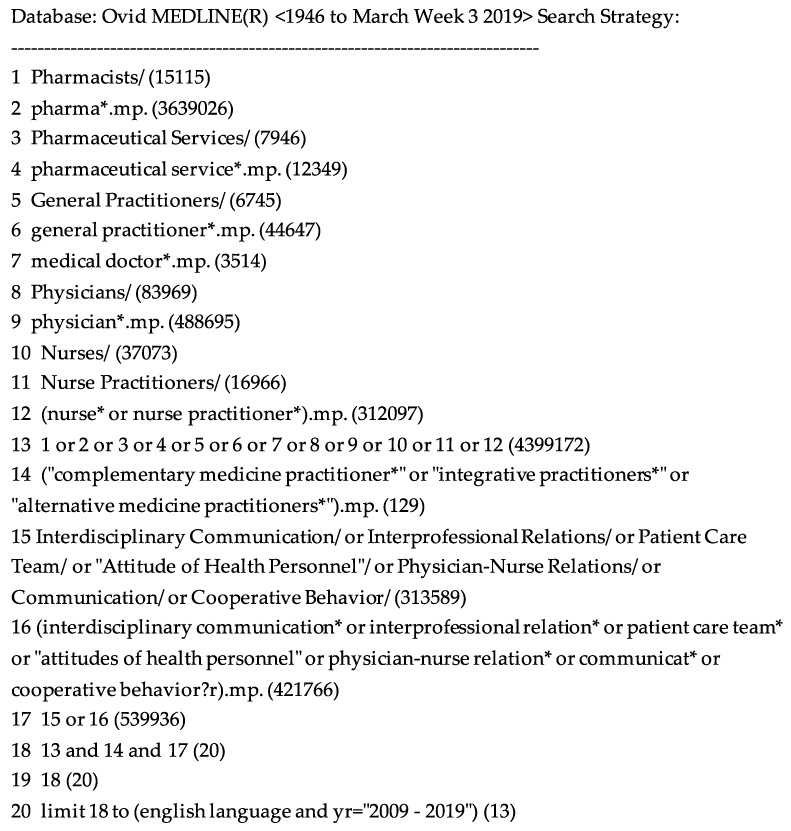
Medline search strategy.

**Figure 2 medicina-55-00650-f002:**
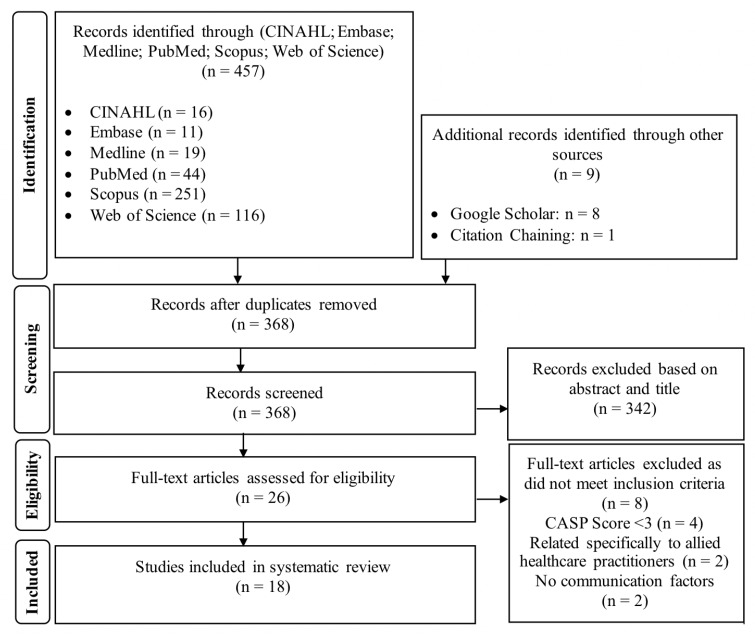
PRISMA flow diagram of literature search method [17].

**Figure 3 medicina-55-00650-f003:**
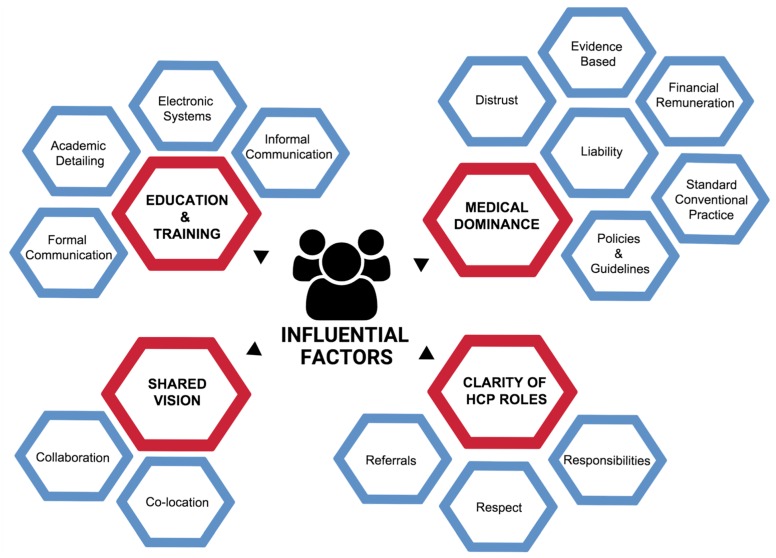
Map of key themes (red) and associated sub-themes (blue).

**Table 1 medicina-55-00650-t001:** Overview of the studies included in this review.

Study	Aims	Method	Sample	Key Findings	CASP Score
**Bollen et al. 2018 [7]**	Identify factors that influence interprofessional collaboration between GP and Pharmacists	Systematic Review, PRISMA	26,452 GP and Pharmacists (37 articles) (International)	Hierarchy between GP and community pharmacists, lack of clarity of roles. Previous experience and co-location may assist	Strong
**Fejzic et al. 2010 [19]**	Investigate HCP views on their relationships and reaching concordant partnerships with consumers regarding use of TCM.	Problem Detection Study Survey/Questionnaire	6 pharmacists 5 TCM-HCP 5 GP (Australia)	3 HCPs agreed on shared information to consumers, IPC through understanding roles, complementary and alternative medicine education required.	Strong
**Foronda et al. 2016 [4]**	Shed light onto what is known about IPC and IPC education.	Integrative Review, Whittemore and Knafl method	Nurses Medical Doctors (18 articles) (Australia)	Biomedical dominance (professional/organisation/structural), lack of trust hindered IPC. Standardised tools, common language and simulations help IPC.	Strong
**Gallagher and Gallagher, 2012 [20]**	Identify factors that help or hinder working relationships between medical doctors and pharmacists.	Narrative review	Pharmacists Medical Doctors (International)	Lack of IPC, trust and perception of autonomy loss determined quality of working partnership. Importance of education and agreed working practices/roles.	Moderate
**Grace and Higgs, 2010 [21]**	Examine the relationships between GP and TCM and their respective roles in co-located medical facilities.	Van Manen Hermeneutic phenomenology, semi structured interviewsFocus groups, In-depth interviews	8 GP 13 TCM-HCP (Australia)	Mutual power sharing and acknowledgement of TCM-HCProles by GP enhance IPC.	Strong
**Gray and Orrock, 2014 [22]**	Explore HCP perspectives on integrating TCM with biomedicine, identify factors influencing referral between TCM-HCP and other HCP	Semi-structured interviews, thematic analysis	2 GP 4 TCM-HCP (Australia)	Informal IPC reinforces collaboration, shared vision and trust. Mutual respect drives IPC. Poor record keeping, and medicolegal risks reduce referral rate	Strong
**Hunter et al. 2012 [23]**	Contribute to research and debate on what constitutes an integrative medicine team and the impact of biomedical dominance	Case study of integrative medicine clinic, mixed methods	6 GP 4 Allied health 4 TCM -HCP 1 Health Service Manager (Australia)	Biomedical dominance, lack of role clarification limited communication and team collaboration	Moderate
**Janamian et al. 2011 [24]**	Develop, implement, evaluate a TCM information resource for General Practice.	Evaluation fact sheets, post-intervention survey	92 GPs (Australia)	Education through factsheets enhance GP knowledge of complementary and alternative medicine, academic detailing, reminder systems and feedback effective, printed materials and lectures least effective.	Strong
**Leach and Tucker, 2017 [25]**	Shed light on the gap between research and TCM practice	Mixed methods cross-sectional study: survey and semi-structured interviews	43 academics 29 academics from TCM journals (International)	Mutual medical language communication encouraged amongst HCP, TCM -HCP have limited research literacy, different TCM ethos	Strong
**McInnes et al. 2015 [1]**	Identify facilitators and barriers influencing collaboration and teamwork between GPs and nurses working in general practice	Integrative Review, thematic analysis	GPs, Nurses 11 articles (International)	Factors impeding collaboration between GP & Nurse: hierarchy, trust, liability, respect	Strong
**O’Reilly et al. 2017 [26]**	Examine accounts and analyse barriers and facilitators of interdisciplinary teamwork in primary care settings from the perspective of service providers	Integrative Review, PRISMA, Normalisation Theory	32 HCP (49 articles) (International)	Traditional hierarchies, remuneration costs, lack of clarity of roles impede IPC. Formal and informal methods of communication may assist.	Strong
**Schadewaldt et al., 2013 [6]**	Summarise the evidence about the views and experiences of nurse practitioners and Medical Doctors with collaborative practice in primary health care settings.	Integrative Review, PRISMA, Whittlemore and Knafl approach, Thematic Synthesis	1641 Medical Doctors 380 Nurse practitioners (27 articles) (Australia)	Lack of role clarification limit IPC. Informal communication, mutual trust and respect, co-location promote IPC.	Strong
**Singer et al. 2013 [27]**	Investigate perspectives of HSM regarding their role in facilitating effective integrative practice between TCM-HCP and other HCP.	Semi-structured interviews;	8 Health Service Managers (Australia)	Facilitators of IPC: meetings, respect, education, shared values, co-location	Strong
**Smith et al. 2014 [28]**	Examine and understand the knowledge and attitudes of clinical health professionals working in reproduction medicine towards evidence-based practice and research.	Cross-sectional online survey	17 Medical Doctors 32 Nurse 22 Counsellor 4 Other (Australia)	Barriers of time and lack of support for evidence base research to improve clinical practice, lack of knowledge of professions and biomedical dominance	Strong
**Soklaridis et al. 2009 [29]**	Explore communication amongst the various stakeholders in an integrative health clinic.	In-depth individual interviews, semi-structured and focus groups	8 Clients 5 Hospital Administrators 8 Board Members 5 Practitioners (Canada)	Hierarchy present between biomedical practitioners and TCM -HCP, lack of formal IPC and referral rates.	Strong
**Supper et al. 2014 [30]**	Identify factors that influence interprofessional collaboration in the primary care setting.	Systematic review, thematic analysis	513 Mental Health Practitioners 374 Allied Health 164 Midwives 58 Pharmacists 48 Receptionists (44 articles) (International)	Hierarchy, lack of clarity of roles, traditional biomedical views impedes IPC. Shared communication tools such as electronic health records may assist.	Strong
**Ung et al. 2017 [31]**	Explore perceptions and opinions of pharmacists and other key stakeholder leaders about TCM barriers and potential solutions.	Semi-structured key informant interviews, pilot explorative interviews	8 Key stakeholders 4 Pharmacists (Australia)	Poor IPC between pharmacists and Medical Doctors about TCM. More TCM education required.	Strong
**Wiese et al. 2010 [32]**	Explore the research and commentary literature on the current and emerging relationship between biomedicine and TCM.	Systematic qualitative review	2011 HCP (55 articles) (International)	Conventional HCP dominate TCM-HCP and prefer selective incorporation of TCM into practice.	Moderate

TCM-HCP = Traditional and Complementary Medicine; GP = General Practitioners; HCP = Health Care Practitioners; IPC = Interprofessional Communication; NP = Nurse Practitioners.

**Table 2 medicina-55-00650-t002:** Summary of thematic analysis: Key themes, subthemes and indicative quotes*.

**1. MEDICAL DOMINANCE**
**1.1 Standard conventional practice (*n* = 11 articles)** [1,7,19,21,22,23,26,29,30,31,32]
**Conventional Health Care Settings** - ‘Biomedical profession gained medical dominance by … establishing occupational boundaries that marginalized its competitors [TCM-HCP]’ [32] - ‘Professional socialisation’ [26] ‘attributes ultimate authority to doctors’ [31] as this is the ‘traditional status of doctors’ and there is the ‘assumption of the GP as the team leader’ [26] - ‘Occupational rivalry’ [22] ‘goes both ways, for example’ [GPs], ‘*you’re certainly not a peer, and the hierarchy is clear*’ [19] - ‘Territorialism around GPs protecting their own professional boundaries and expertise’ [1] and GPs often perceived interprofessional ‘initiatives as a threat to their autonomy’ [22] - ‘Tendency for biomedical doctors to control patient care and use biomedical language as the main form of communication’ [23] - Between doctors and pharmacists there was ‘bidirectional ambivalence, lack of communication and lack of cooperation’ [31] **Complementary and Integrative Health Care Settings** - ‘Disapproval of [TCM-HCP] by doctors’ [31] - ‘TCM and biomedical relationship remain distant’. There is ‘increasing acceptance by biomedical practitioners of TCM therapies but little interest in developing a working relationship’. There is a ‘perception of bleak future’ where ‘TCM integration will lead to biomedical dominance of [the TCM-HCP] profession’. Some TCM-HCP are ‘willing to accept inclusion in mainstream healthcare under ‘selective incorporation’ model favoured by the biomedical profession’ whereas others think that ‘legitimacy and status [in mainstream healthcare] may cost, with [TCM] becoming a marginalized sub-field’ resulting in a ‘loss of occupational autonomy’ where ‘traditional philosophies are removed in favour of biomedical philosophy as the basis of practice’ [32] - ‘Non-inclusive practice style results in the dilution of [TCM-HCP] role in the health care system’ [21] - ‘Allied-health and TCM-HCP would prefer a less hierarchal system’ [23] - ‘Fraught with power struggles and entrenched in medical hierarchy’ between [**complementary and alternative medicine**] practitioners and biomedical practitioner’ [29]
**1.2 Distrust (*n =* 7 articles)** [1,6,7,19,20,26,32]
- ‘Took steps to gain [trust of doctors]’ [26] - ‘GP Distrust in nurse’s knowledge and skills to perform competently’ [1] - ‘Medical doctors are not comfortable with their patients seeing TCM-HCP’ [19] - ‘Biomedical practitioners … less likely to endorse a patient’s suggestion to refer to a TCM-HCP’ [32] - ‘Trustworthiness (i.e., Pharmacist performance), role specification and initiating behavior … dictated quality of the doctor-pharmacist working relationship’ [20] - Physicians perceived nurses ‘inattentiveness … unwillingness to discuss goals of care and feelings that list of signs and symptoms had to be provided instead of just stating…clinical problem’ [4] - ‘Physicians concern about pharmacist’s capabilities … GPs perceived TCM-HCPs having lack of knowledge and skills regarding (increased) patient care’ [7]
**1.3 Liability and Confidentiality (*n =* 5 articles)** [1,4,22,26,30]
- ‘GPs … cognizant of potential legal implications created by the autonomous practice of nurses and the subsequent exposure … to degree of risk’ [1] - ‘Pharmacists’ medico-legal responsibility placed limits on the extension of their roles to diagnosis and prescription’ [30] - ‘Data confidentiality … perceived as significant when medical data are shared with pharmacists’ [30] - ‘Medico-legal concerns surrounding safety and duty of care of a referral’ [22]
**1.4 Evidence Base (*n =* 3 articles)** [25,28,31]
- ‘Lack of evidence for efficacy for research skills … and safety of TCM-HCP and lack of access to trustworthy information and support’ [31] - ‘Translation [TCM-HCP] … findings not being written in a way that is intelligible, useful or usable to practitioners’ [25] - ‘Tension existing between interest and support for evidence-based research and recognizing its value on one hand and experiencing barriers of time and lack of support’ [28]
**1.5 Financial Remuneration (*n =* 4 articles)** [6,25,28,32]
- ‘Health care financing has contributed to the mainstream acceptance of TCM’ [32] - ‘Incentives … lack to be engaged in research training and evidence-based training’ [25] - ‘Lack of resources to undertake research and lack of organizational support’ [28] - ‘Economic constraints … health care system did not sufficiently reimburse NP services’ [6]
**1.6 Policies and Guidelines (*n =* 3 articles)** [19,26,30]
- ‘Enhancing communication between all stakeholders in healthcare should be the responsibility of policy makers and health professionals’ [19] - ‘Structured and disseminated disease-specific management guidelines, with information about **complementary and alternative medicine****s**’ [30] - ‘Guidelines contain confirmed interactions between commonly used TCM products … and relevant conventional medications, known side effects … general advice to health professionals on how to discuss **complementary and alternative medicine**-related issues … referrals to **complementary and alternative medicine** practitioner’ [30] - ‘Clear policies … and clear focus on patient care’ [26]
**2. Clarity OF HCP ROLES** **2.1 Respect (*n =* 4 articles)** [1,7,21,27]
- ‘GPs had misconceptions and lacked understanding of pharmacy services’ [7] - ‘Acknowledgement organisationally of the particular expertise’ [27] - ‘Professionalism is founded on concept of respect for other practitioners’ [21] - ‘Confidence in professional competence underpinned trust and respect’ [1] - ‘Consumers choice to use **complementary and alternative medicine** is not respected by all health practitioners involved in their care’ [1]
**2.2 Responsibilities (*n =* 5 articles)** [1,6,7,23,30]
- ‘Nurses have poor attendance at practice meetings … limited opportunities’ [1] - ‘Substitution of doctors by nurse practitioners is constrained by difficulties in acquiring the new skills needed to address multidimensional consultations’ [30] - ‘More contact with the referring doctor, the more they [GPs] realise that allied health practitioners play an integral role in management of their patients’ [26] - ‘Medical practitioners reported losing control about patient triage through introduction of NPs’ [6] - ‘Practitioners not having an in-depth understanding of each other’s modalities’ [23] - ‘Lack of definition, awareness and recognition of the role of each professional’ [6] - ‘Dealing with multiple professions, increase duplication of tasks, costs and fragmented healthcare’ [7]
**2.3 Referrals (*n =* 2 articles)** [21,27]
- ‘Lack of time available to allied health care professionals enables cross referral to appropriate healthcare services’ [27]
**3. SHARED VISION** **3.1 Co-location (*n =* 2 articles)** [27,30]
- ‘More opportunities for case discussion among different integrative healthcare practitioners’ [27] - ‘Shared location with a meeting space and dedicated to collaboration is needed’ [30] - ‘Structural facilitating factors are shared facilities and organization’ [30]
**3.2 Collaboration and Partnerships (*n =* 11 articles)** [1,7,19,20,21,22,25,26,27,31]
- ‘[GP] Concerns about collaborating … reduction in GP income’ [26] - Some ‘doctors viewed nurses in general practice as resource and complementary to their services’ [1] - ‘Agreed working practices, attitudes to patient care and cultural differences between respective professions [GPs, pharmacists]’ [20] - ‘Establishing avenues for cross-disciplinary education, good communication practice’ [27] - ‘Harmonious partnerships through understanding individual’s agendas’ [19] - ‘Must not be obstinate and think that our own discipline is more important or less important’ [21] - ‘Collaborative information exchange model … whereby knowledge producers and knowledge users work closely together to overcome barriers’ [25]
**3.3 Shared Decision Making (*n =* 6 articles)** [1,21,22,23,27,30]
- ‘Case information was shared, and treatment goals were developed cooperatively’ [27] - ‘Sharing a philosophy of care and a common understanding pertaining to scope of practice and area of expertise’ [22] - ‘Agreement among the practitioners of a shared vision, open-minded culture, credible supporters, suitable facilities and confidence in the clinical competency of the other practitioners’ [23]
**4. EDUCATION AND TRAINING** **4.1 Formal Communication (*n =* 6 articles)** [1,4,24,26,27,31]
- ‘Combined supervision groups, case review/ conferences, team meetings, staff meetings and debriefing’ [27] - ‘Practice meetings give opportunities for disciplines to share decision making, goal setting and responsibilities’ [1] - ‘Functions of structures for formal clinical meetings, dedicated events or initiatives to support teams or formal appraisal process’ [26] - ‘SBAR intervention to improve communication in a tertiary care centre … nurses claimed the tool has eliminated errors due to assumptions’ [4] - ‘Academic detailing and use of local opinion leaders were the most effective techniques in changing physician performance’ [24] - ‘Physician reminder systems … audit and feedback techniques marginally effective and conferences and printed materials were the least effective’ [24] - ‘Process of formal evaluation … helpful for enabling and supporting team working and development’ [31] - ‘66% of GPs agreed that prior to receiving the fact sheets [of TCMs] they felt they did not have adequate knowledge to discuss those herbal medicine options’ [24]
**4.2 Informal Communication (*n =* 4 articles)** [4,7,22,26]
- ‘Practitioners … drive the integration, with shared files via clinic intranet, structured case-based meetings, and informal corridor and lunchroom chats’ [22] - Nurses: ‘disorganiz[ed] information, illogical flow of content, lack of preparation to answer questions, inclusion of extraneous or irrelevant information and delay in getting to the point’ [4] - ‘Ad hoc interactions … generally described as positive and effective for shared decision making and informational continuity of care for patients’ [26] - ‘Inaccessibility such as difficulties getting access to GP, community pharmacists or patients medical records and disorganised charting systems’ [7] - ‘Regular telephone or face to face contact between the two professionals, ensuing community pharmacists received feedback from the GP … regular and proactive communication and information sharing’ [7]
**4.3 Electronic System (*n =* 4 articles)** [6,23,26,30]
- ‘Technical challenges with electronic messaging systems’ [23] and there was a ‘lack of face-to-face contact with other practitioners’ [23] - ‘[digital versatile discs] explaining the education pathway and the skills of NPs increased significantly the knowledge of primary case medical practitioners and their positive attitude towards NP’ [6] - ‘[Information Technology]’ systems and the use of Electronical Medical Records as well as electronic patient booking systems’ [26]
**4.4 Knowledge and Educational Sessions (*n =* 12 articles)** [1,6,19,20,21,25,26,27,30,31,32]
- There was a ‘lack of awareness by medical practitioners of the scope of practice of NPs, their level of education and what is inherent to their role’ [6] and hence many ‘doctors strongly believed that the education of nurses did not support their role as autonomous clinicians’ [1] - Doctors receiving training from pharmacists’ … felt more confident in conducting medication review’ [20] - ‘Encouraging academics/scientists to communicate to practitioners in a language that they would understand’ [25] - ‘Education on benefits, risks and marketing regulations on **complementary and alternative medicine****s** should be directed to … health practitioners to improve their therapeutic judgement’ [19]

* According to Braun and Clark’s Methodology [16].

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
