# Peer review of "Conventional and Complementary Medicine Health Care Practitioners’ Perspectives on Interprofessional Communication: A Qualitative Rapid Review"

_medicina, 2019, doi:10.3390/medicina55100650_

Round 1

Reviewer 1 Report

Thank you for the work you have done on this paper. Really interesting topic and analysis. And thank you for your careful preparation of the manuscript--your excellent grammar and editing made the manuscript a delight to read. 

The purpose of this project was to investigate and synthesize the current literature that reports conventional and traditional & complementary medicine healthcare provider experiences with, and perceptions of interprofessional communication, and identify factors that influence interprofessional communication, with a focus on Australia. 

I have a few questions:

I am unclear on why the focus is Australia. Was that limitation necessary to make the review a manageable size? How many additional studies would be included if the Australian limitation was not imposed? I can see why you might want that limitation for the purposes of your local clinical work, but I don't see the value of that limitation for knowledge generation in the literature. Please include a citation to document the 2nd sentence of the paper (p. 2, lines 41-43) regarding reduced morbidity and costs and improved well-being and satisfaction. I would appreciate a sentence or two describing the “rapid review approach” so I don’t need to look up the reference about it. Similarly, I would appreciate a brief description of the CASP checklist and what topics are included in the three level scale. What do “yes scores” mean? You mention the interprofessional perspective of the authors/reviewers in the limitations section (p. 17, lines 301-303). That information would be helpful in the methods section instead, where you describe the screening and review process. When you mention “the authors’” on p. 3, line 95, do you mean the authors of the articles you are reviewing? Or the authors of the current article? Perhaps restate that sentence—I am not sure what you mean so I don’t have a suggestion on how to reword it. Similarly, on p. 3, line 100, I am not sure what the phrase “allowing common arrangements from the array of studies with a focus on exploring commonalities …..” means. 4, line 137, who were the “other HCP?” What professions did they represent? 4, line 138, I am not sure why it makes sense to review other literature reviews within a literature review paper. Why don’t you use those literature reviews to identify the original papers and include those in your review? Could you include a rationale somewhere about why it is appropriate to include literature reviews in a literature review? In Table 1, I am assuming that you directly copied/pasted the aims from each of the articles. Is that correct? Did you also directly copy/paste the “key findings?” If not, how much leeway was used in paraphrasing the key findings? How is validity/accuracy of those key findings (as reported in this table) assured? In figure 2, “trust” is one of the factors included under “Medical Dominance.” From the description, sounds like you are actually talking about “distrust” not “trust.” Table 2, Are those themes all direct quotes from the articles that are cited? Or are they paraphrased? Perhaps comment on whether or not they are direct quotes when introducing Table 2 in the body of the paper. Many acronyms and abbreviations are used throughout the paper, which makes the paper difficult to read/follow. Would prefer to delete any abbreviations that are not absolutely necessary. Also please be sure that all abbreviations are spelled out when they are first used (AHPs? NPs? CP? MP? DVDS? IT? CAMs?) The term “allied health” is used—could you please define what you mean by that term? I think it is used slightly differently in different parts of the world. This paper is described as a “qualitative literature review.” What does that mean? Are you only reviewing qualitative literature? If so, that should be specified in the methods. page 15, lines 187-188—I think the sentence may be mis-stated with cause & effect the opposite way. Are you trying to say that “The reluctance of medical doctors to disclose their rationale for treatment approaches to other HCP is due to their position as team leader?” Please clarify that statement, I am not sure I understood it as written. page 17, lines 304-305. Why did you choose to use manual extraction and coding of themes rather than using a qualitative analysis program, especially when it may have introduced errors and miscoding?

I hope to see a revision of this paper. It is a very interesting topic. Thank you for your work on it.

Author Response

Dear Reviewer 1,

Thank you for your valuable feedback and contribution to improving our paper.

Please find our point by point response in the attached letter of response.

In appreciation,

The authorship team

Reviewer 2 Report

Review of the manuscript ID: medicina-585552 Conventional and Complementary Medicine Practitioners' perspectives on interprofessional communication; A qualitative rapid review

Introduction

The introduction appears as logical and stringent. The aim of the study seems relevant and valuable from a scientific and clinical point of view. Please, clarify the choise of country/setting, why Australia?

Methods

Please, explain why the time limit of ten years was chosen when including articles and also state the specific time span in this section, is it 2019 – 2009?

Results

It would be more logical to gather all the background information about the data collection (Figure 1 and Table 1) in the method section. Thereafter, the reader expects to get information about the data analysis as described by Braun and Clark. Please clarify with one or two examples how the specific data in this study was analysed. Right now, the analysis description is quite theoretical and isolated described disconnected from the actual data set.

The result section should start at page 150, line 145 with the sentence “Four key themes were identified: 1)…”

The results seem credible and reasonable. However, 20 sub-themes appear as a bit too much due to the size of the data set. When going through the themes and sub-themes, I think it would have been possible to condense some of the sub-themes which are similar to each other and overlap.

How was the credibility of the results secured among the authors, in more detail?

Discussion

Pleas look at page 15, line 163 and 169. Could you use another term the second time instead of repeating “to the best of our knowledge”. I also think it would be of benefit to write a short preamble as part of the first paragraph as a service to the reader, what will be discussed in this section.

4.1 Medical dominance – a very important aspect is in focus here that needs to be highlighted even more in society in general. The medical doctors always have had a dominant position in society as well as in the health care area. This must change in the future if the patients are going to be cared for in a holistic way. I’m afraid that guidelines and policies are not enough, it’s a question of macro structures, power and privileges at the expense of other professionals’ knowledge and autonomy.

Author Response

(The authors gave the same response as above.)
